# Improved Parsimonious Topic Modeling Based on the Bayesian Information Criterion

**DOI:** 10.3390/e22030326

**Published:** 2020-03-12

**Authors:** Hang Wang, David Miller

**Affiliations:** Electrical Engineering and Computer Science Department, The Pennsylvania State University, State College, PA 16802, USA; hzw81@psu.edu

**Keywords:** topic model, Bayesian information criterion, expectation maximization algorithm, medical abstracts

## Abstract

In a previous work, a parsimonious topic model (PTM) was proposed for text corpora. In that work, unlike LDA, the modeling determined a subset of salient words for each topic, with topic-specific probabilities, with the rest of the words in the dictionary explained by a universal shared model. Further, in LDA all topics are in principle present in every document. In contrast, PTM gives sparse topic representation, determining the (small) subset of relevant topics for each document. A customized Bayesian information criterion (BIC) was derived, balancing model complexity and goodness of fit, with the BIC minimized to jointly determine the entire model—the topic-specific words, document-specific topics, all model parameter values, and the total number of topics—in a wholly unsupervised fashion. In the present work, several important modeling and algorithm (parameter learning) extensions of PTM are proposed. First, we modify the BIC objective function using a lossless coding scheme with low modeling cost for describing words that are non-salient for *all* topics—such words are essentially identified as wholly noisy/uninformative. This approach increases the PTM’s model sparsity, which also allows model selection of more topics and with lower BIC cost than the original PTM. Second, in the original PTM model learning strategy, word switches were updated sequentially, which is myopic and susceptible to finding poor locally optimal solutions. Here, instead, we jointly optimize all the switches that correspond to the same word (across topics). This approach jointly optimizes many more parameters at each step than the original PTM, which in principle should be less susceptible to finding poor local minima. Results on several document data sets show that our proposed method outperformed the original PTM model with respect to multiple performance measures, and gave a sparser topic model representation than the original PTM.

## 1. Introduction

Topic modeling [1] is a type of statistical modeling for finding the “topics” that occur in a collection of documents. The latent Dirichlet allocation (LDA) [2] is one of the well-known topic models. The LDA topic model assumes that each topic is a probability mass function defined over the given vocabulary, and for each of the documents, every word follows a document specific mixture over the topics. In an improvement of LDA called parsimonious topic modeling (PTM) [3], two shortcomings of LDA are discussed. First, all words have their own probability parameters under every topic in LDA; this strategy uses a huge number of parameters, with the model potentially prone to overfitting. Second, in LDA, every topic is assumed to be present in every document, with a non-zero probability. However, PTM gives a sparser description in the two aspects mentioned above. First, some words are not topic specific for certain topics—they use a universal shared model. Second, in each document, only some of the topics occur with a non-zero probability.

The Bayesian information criterion (BIC) [4,5] is a widely used criterion for model selection. There are two parts in the negative logarithm of the Bayesian marginal likelihood: the likelihood of the data and the model complexity cost. So, we can use BIC to balance data fitting goodness and model complexity. The BIC cost function derived for PTM [3] improves over “vanilla” BIC in two aspects. First, the proposed BIC has differentiated cost terms based on different effective sample sizes for different types of parameters; second, PTM introduces a shared feature representation to decrease the effective feature dimensionality of a topic. Our contribution here over PTM [3] is that we give a cheaper expression to describe wholly uninformative words, which thus encourages an even sparser model. Inspired by the intuition that in text corpora there are a large proportion of words that are not related to any topics, we implemented an optimization method to jointly optimize all the parameters related to a single word under each topic, to encourage the possibility that all topics choose to use a shared (uninformative) representation of a given word. Our work improves on the PTM model in encouraging a sparser and more reasonable representation of topics.

Our model solves both the unsupervised feature selection and model order (number of topics) selection problems. First we fix the hyper-parameter (number of topics) to a large number, and we get an optimized structure of the model by determining the topic-specific words under each topic and which topics occur in each document. Then, we change (reduce) the number of topics (hyper-parameter) by removing some topics. For each value of this hyper-parameter we train the model and compute BIC, and the optimized hyper-parameter (chosen model order) is the one with the lowest BIC value.

## 2. Notation

Suppose a corpus consists of *D* documents and *N* unique words, with d∈{1,2,…,D} and n∈{1,2,…,N} the document and word indices, respectively, and with unique topic indexed by j∈{1,2,…,M}, *M* being the total number of topics (model order). Here are some different definitions used in this paper:

$L_{d}$ is the number of unique words in document *d*.

$w_{id}\in \left\{1,2,\ldots ,N\right\}$, $i=1,\ldots ,L_{d}$ is the *i*-th word in document *d*.

$v_{jd}\in \left\{0,1\right\}$ is the topic “switch”; it indicates whether topic *j* is present in document *d*.

Topic *j* is present in document *d* if $v_{jd}=1$; otherwise $v_{jd}=0$.

$M_{d}\equiv \sum_{j=1}^{M}v_{jd}\in \left\{1,2,\ldots M\right\}$ is the number of topics present in document *d*.

$\alpha_{jd}$ is the proportion for topic *j* in document *d*.

In each topic:

$\beta_{jn}$ is the topic-specific probability of word *n* under topic *j*.

$\beta_{0n}$ is the shared probability of word *n*.

$u_{jn}\in \left\{0,1\right\}$ indicates whether ($u_{jn}=1$) or not ($u_{jn}=0$) word *n* is topic-specific under topic *j*.

$Nj\equiv \sum_{n=1}^{N}u_{jn}$ is the total number of topic-specific words under topic *j*.

${\bar{L}}_{j}\equiv \sum_{n=1}^{N}L_{d}v_{jd}$ is the sum of the length of the documents for which topic *j* is present.

## 3. Methodology

### 3.1. “Bag of Words” Model

A bag of words model [6] is commonly used in document classification where the count of each word is used as a feature for class discrimination. Using the bag of words model, we can transform the text corpora into a feature matrix, where each row vector $x=(x_{1},x_{2},\ldots ,x_{D})$ in the matrix is a bag for each document, with the length of the vector the total number of unique words in the whole text corpora. In the row vector, each position represents a single word, and the value in that position is the number of times this word occurs in the document.

### 3.2. Parsimonious Topic Model (PTM)

We first introduce PTM’s data generation method.

For each document d=1,2,…,DFor each word $i=1,2,\ldots ,L_{d}$

Randomly select a topic based on the probability mass function (pmf) $\left\{\alpha_{jd}v_{jd},j=1,2,\ldots ,M\right\}$.Given the selected topic *j*, randomly generate the *i*-th word based on the topic’s pmf over the word space $\left\{\beta_{jn}^{u_{jn}}\beta_{0n}^{1-u_{jn}},n=1,2,\ldots ,N\right\}$

Here $v_{jd}$ is the topic switch that indicates whether topic *j* is present in document *d*. If $v_{jd}=1$, it means that topic *j* is present in document *d* and $\alpha_{jd}$ is treated as a model parameter. $\beta_{jn}$ and $\beta_{0n}$ are the topic-specific probability of word *n* under topic *j* and the shared probability of word *n*, respectively.

Based on the above data generation, we can get the data likelihood of a document dataset χ under our model (H,Θ):(1)$$p\left(\chi |H,\Theta \right)=\prod_{d=1}^{D}\prod_{i=1}^{L_{D}}\prod_{j=1}^{M}\left[\alpha_{jd}v_{jd}\beta_{jn}^{u_{jw_{id}}}\beta_{0n}^{1-u_{jw_{id}}}\right].$$

Here $u_{jn}$ is the word switch that indicates if word *n* is topic-specific under topic *j*. The model *structure* parameters, denoted by H{v,u,M}, consist of two kinds of switches and the number of topics, *M* (model order). Likewise, the model parameters, given a fixed model structure, are denoted by $\Theta =\{\left\{\alpha_{j}\right\},\left\{\beta_{jn}\right\},\left\{\beta_{0n}\right\}\}$. The model structure together with the model parameters constitutes the PTM model. In PTM the parameters are constrained by the following two conditions:

First, $\alpha_{jd}$ is the probability that topic *j* is present in document *d*, and $v_{jd}$ determines whether or not topic *j* is present. The probability mass function must sum to one. So, we have:(2)$$\sum_{j=1}^{M}\alpha_{jd}v_{jd}=1,\forall d.$$

Additionally, the word probability parameters $\left\{\beta_{jn},n=1,\ldots ,N\right\}$ and $\left\{\beta_{0n},n=1,\ldots ,N\right\}$ must satisfy a pmf constraint for each topic:(3)$$\sum_{n=1}^{N}\left(u_{jn}\beta_{jn}+\left(1-u_{jn}\right)\beta_{0n}\right)=1,\forall j.$$

Based on the PTM described above, we must determine the model parameters Θ and the model structure, *H*. Assuming the model structure is known, we can estimate the model parameters using the expectation maximization (EM) algorithm. By introducing, as hidden data, random variables that indicate which topic generates each word in each document, we can compute the expected complete data log likelihood and maximize it subject to the two constraints mentioned above. The model selection is more complicated. We need to derive a BIC cost function to balance the model complexity and the data likelihood. For the PTM model a generalized expectation maximization (GEM) [7,8] algorithm was proposed to update the model parameters (Θ) and the model structure (H) iteratively. In the following section, we give a derivation of BIC and the GEM algorithm for our modified PTM model.

### 3.3. Derivation of PTM-Customized Bayesian Information Criterion (BIC)

In this section we derive our BIC objective function, which generalizes the PTM BIC objective [3]. A naive BIC objective has the following form:(4)$$BIC=K\log\left(n\right)-2\log\left(\hat{L}\right).$$

Here $\hat{L}$ is the maximized value of the data likelihood of the model with structure *H*, that is, $\hat{L}=p\left(D|H,\hat{\Theta }\right)$), where $\hat{\Theta }$ is the collection of parameters that maximize the likelihood function.

Additionally, *n* is the number of data points (documents) in the dataset χ and *K* is the number of free parameters in the model.

However, the Laplace approximation used in deriving this BIC form is only valid under the assumption that the feature space is far smaller than the sample size, and for our topic model, the feature space (word dictionary) is quite large in practice. Moreover, in the naive BIC form, all the parameters incur the same description length penalty (the log of the sample size), but in PTM different types of parameters contribute unequally to the model complexity. So, a new customized BIC is derived for PTM.

The Bayesian approach to model selection is to maximize the posterior probability of the model *H* given the dataset χ. When applying Bayes’ theorem to calculate the posterior probability, we get:(5)$$p\left(H|\chi \right)=\frac{p(\chi |H)p(H)}{p(\chi )}.$$

Here we define:(6)I=p(χ|H)=∫p(χ|H,Θ)p(Θ|H)dΘ,
where p(Θ|H) is the prior distribution of the parameters given the model structure *H*. Then, we need to use Laplace’s method to approximate *I*, given the knowledge that for large sample size p(χ|H,Θ)p(Θ|H) peaks around the maximum point (posterior mode $\hat{\Theta }$). We can rewrite *I* as:(7)I=p(χ|H)=∫exp(log(p(χ|H,Θ)p(Θ|H)))dΘ.

We can now expand log(p(χ|H,Θ)p(Θ|H)) around the posterior mode $\hat{\Theta }$ using a Taylor series expansion.
(8)$$log\left(p\left(\chi |H,\Theta \right)p\left(\Theta |H\right)\right)\approx log\left(p\left(\chi |H,\hat{\Theta }\right)p\left(\hat{\Theta }|H\right)\right)+\left(\Theta -\hat{\Theta }\right)\nabla_{\Theta }{Q|}_{\hat{\Theta }}-\frac{1}{2}{\left(\Theta -\hat{\Theta }\right)}^{T}{\tilde{\Sigma }}_{\Theta }\left(\Theta -\hat{\Theta }\right),$$
where Q≡log(p(χ|H,Θ)p(Θ|H)) and ${\tilde{\Sigma }}_{\Theta }\equiv -\Sigma_{\Theta }$, where $\Sigma_{\Theta }$ is the Hessian matrix (i.e., $\Sigma_{i,j}=\partial^{2}\frac{Q}{\partial \Theta_{i}\partial \Theta_{j}}{|}_{\hat{\Theta }}$).

Since *Q* attains its maximum at $\hat{\Theta }$, $\nabla_{\Theta }{Q|}_{\hat{\Theta }}=0$ and ${\tilde{\Sigma }}_{\Theta }\equiv -\Sigma_{\Theta }$ is negative definite. We can thus approximate *I* as follows:(9)$$I=p\left(\chi |H\right)\approx p\left(\chi |H,\hat{\Theta }\right)p\left(\hat{\Theta }|H\right)\int e^{\frac{1}{2}{\left(\Theta -\hat{\Theta }\right)}^{T}{\tilde{\Sigma }}_{\Theta }\left(\Theta -\hat{\Theta }\right)}.$$

With the above form of the approximation, $e^{\frac{1}{2}{\left(\Theta -\hat{\Theta }\right)}^{T}\tilde{\Sigma_{\Theta }}\left(\Theta -\hat{\Theta }\right)}$ is a scaled Gaussian distribution with mean $\hat{\Theta }$ and covariance ${\tilde{\Sigma }}_{\Theta }$. Thus:(10)$$\int e^{\frac{1}{2}{\left(\Theta -\hat{\Theta }\right)}^{T}\tilde{\Sigma_{\Theta }}\left(\Theta -\hat{\Theta }\right)}={\left(2\pi \right)}^{\frac{k}{2}}{\left|{\tilde{\Sigma }}_{\Theta }\right|}^{-\frac{1}{2}},$$
where *k* is the number of parameters in Θ. So we have the approximation of *I*:(11)$$I=p\left(\chi |H\right)\approx p\left(\chi |H,\hat{\Theta }\right)p\left(\hat{\Theta }|H\right){\left(2\pi \right)}^{\frac{k}{2}}{\left|{\tilde{\Sigma }}_{\Theta }\right|}^{-\frac{1}{2}}.$$

BIC is the negative log model posterior:(12)$$BIC=-log\left(\hat{I}p\left(H\right)\right)\approx \frac{k}{2}log\left(2\pi \right)+\frac{1}{2}log(|{\tilde{\Sigma }}_{\Theta }\left|\right)-log\left(p\left(\chi |H,\hat{\Theta }\right)\right)-log\left(p\left(\hat{\Theta }|H\right)\right)-log\left(p\left(H\right)\right).$$

Note that $p(\hat{\Theta }|H)$, the prior of the parameters given the structure *H* can be assumed to be a uniform distribution (i.e., a constant). So, this term can be neglected. The $log(p\left(\chi |H,\hat{\Theta }\right))$ term is the data likelihood, and *k* is the total number of model parameters. Now we need to approximately calculate $\frac{1}{2}log(|{\tilde{\Sigma }}_{\Theta }\left|\right)$ and log(p(H)).

To do so, we assume that ${\tilde{\Sigma }}_{\Theta }$ is a diagonal matrix. We thus obtain:(13)$$\frac{1}{2}log(|{\tilde{\Sigma }}_{\Theta }\left|\right)\approx \frac{1}{2}\sum_{d=1}^{D}\left(M_{d}-1\right)log\left(L_{d}\right)+\frac{1}{2}\sum_{j=1}^{M}\sum_{d=1}^{D}u_{jd}log\left({\bar{L}}_{j}\right)+\frac{1}{2}\sum_{d=1}^{D}log\left(\sum_{j=1}^{M}{\bar{L}}_{J}\right).$$

The terms on the right represent the cost of the model parameters $\left\{\alpha_{jd}\right\}$, $\left\{\beta_{jn}\right\}$, and $\left\{\beta_{0n}\right\}$, respectively. Note that in the naive BIC form, each parameter pays the same cost $\frac{1}{2}log\left(samplesize\right)$. Here we instead use the effective sample size. The effective sample size of $\alpha_{hd}$ is $L_{d}$, parameter $\beta_{jn}$ has sample size ${\bar{L}}_{j}$, and the parameter $\beta_{0n}$ has sample size $\sum_{j=1}^{M}{\bar{L}}_{j}$.

Another term to be estimated is log(p(H))=log(p(v))+log(p(u)). For log(p(v)), in each document *d*, suppose the number of topics follows a uniform distribution, and the switch configuration also follows a uniform distribution over all MMd switch configurations. We then obtain:(14)−log(p(v))=Dlog(M)+∑d=1DlogMMd.

For log(p(u)) we propose here a probability model that can jointly estimate log(p(u)) and the corresponding parameter cost of $\beta_{jn}$, $\beta_{0n}$. For each word *n*, we define three types of configurations of the word switches $\left\{u_{jn},j=1,\ldots ,M\right\}$: (1) each word is topic-specific (i.e., $\sum_{j=1}^{M}u_{jn}=M$); (2) all the words are not topic-specific (i.e., $\sum_{j=1}^{M}u_{jn}=0$); (3) some, but not all, components use the shared distribution (i.e., $0<\sum_{j=1}^{M}u_{jn}<M$). For cases 1 and 2, there is only one possible configuration of the word switches related to a word (all *open* or all *closed*), so the probability associated with this configuration is 1; for case 3 there are $2^{M}$ possible configurations. Assuming these are equally likely under this case log(p[u])=Mlog2. We can then estimate −log(p(u)) plus the two terms $\sum_{j=1}^{M}\sum_{d=1}^{D}u_{jd}log\left({\bar{L}}_{j}\right)+\sum_{d=1}^{D}log\left(\sum_{j=1}^{J}{\bar{L}}_{J}\right)$ in $log(|{\tilde{\Sigma }}_{\Theta }|)$. That is, we have
(15)$$\begin{array}{cc} -log\left(p\left(u\right)\right)+\frac{1}{2}\sum_{j=1}^{M}\sum_{d=1}^{D}u_{jd}log\left({\bar{L}}_{j}\right)+\frac{1}{2}\sum_{d=1}^{D}log\left(\sum_{j=1}^{M}{\bar{L}}_{j}\right)\approx & \sum_{n=1}^{N}(\frac{F_{1}\left(u_{n}\right)}{2}log\left(\sum_{j=1}^{M}{\bar{L}}_{j}\right)+\frac{F_{2}\left(u_{n}\right)}{2}\sum_{j=1}^{M}log\left({\bar{L}}_{j}\right)\\ +F_{3}\left(u_{n}\right)\left(\frac{1}{2}log\left(\sum_{j=1}^{M}{\bar{L}}_{j}\right)+\frac{1}{2}\sum_{j=1}^{M}u_{jn}log\left({\bar{L}}_{j}\right)\right). \end{array}$$

Here, $F_{1}\left(u_{n}\right)=\left\{\begin{array}{c} \;1,\;if\;\sum_{j=1}^{M}u_{jn}=0\\ \;0,\;otherwise \end{array}\right.$, $F_{2}\left(u_{n}\right)=\left\{\begin{array}{c} \;1,\;if\;\sum_{j=1}^{M}u_{jn}=M\\ \;0,\;otherwise \end{array}\right.$, $F_{3}\left(u_{n}\right)=\left\{\begin{array}{c} \;1,\;if\;0<\sum_{j=1}^{M}u_{jn}<M\\ \;0,\;otherwise \end{array}\right.$.

Based on the derivation above, the BIC cost function for our modified PTM model is:(16)BIC=Dlog(M)+∑d=1DlogMMd+12∑d=1D(Md−1)log(Ld2π)−log(p(D|H,Θ))∑n=1N(F1(un)2log(∑j=1ML¯j2π)+F2(un)2∑j=1Mlog(L¯j2π)+F3(un)(12log(∑j=1ML¯j2π)+12∑j=1Mujnlog(L¯j2π)).

### 3.4. Generalized Expectation Maximization (EM) Algorithm

The EM [9] algorithm is a popular method for maximizing the data log-likelihood. For unsupervised learning tasks, we only have the data points χ, but we do not have any “labels” for those data points, so during the maximum likelihood estimation (MLE) process we introduce “label” random variables which are called latent variables. The EM algorithm can be described as follows:

E-step: With the parameters fixed, we compute the expectation of the latent variables p(Z|χ,Θ), which gives the class information for each data point. Using the expectation of the latent variables we can compute the expectation of the complete data log-likelihood:(17)$$E\left[L_{c}\right]=\sum_{Z}p\left(Z|D,\Theta \right)log(p\left(D,Z|\Theta \right).$$M-step: We update the parameters Θ to find the maximum value of the expected complete data log-likelihood. By doing the E-step and M-step iteratively, the expected complete data log-likelihood strictly increases and typically converges to a local optimum of the *incomplete* data log-likelihood (or of the expected BIC cost function). However, note that in PTM there are not only the model parameters Θ but also the model structure *H* over which we need to optimize. The original EM algorithm cannot be applied here because one cannot get jointly optimal closed form estimates of both the model parameters Θ and the structure parameters *H* to maximize $E[L_{c}]$. However, a generalized expectation maximization (GEM) [7,8] algorithm is proposed, which alternately jointly optimizes $E[L_{c}]$ over Θ and then over *subsets* of the structure parameters, *H* given fixed Θ.

Our GEM algorithm is specified for fixed model order, M. First we introduce the hidden data *Z*: $Z_{id}$ is an M-dimensional binary vector, with a “1” indicating the topic of origin for the word $w_{id}$. For example, if the element $Z_{id}^{\left(j\right)}=1$ and other elements of $Z_{id}$ are all equal to zero, topic *j* is the topic of origin for word $w_{id}$.

Our GEM algorithm strictly descends in the BIC cost function Equation (Equation 16). It consists of an E-step followed by an M-step that minimizes the expected BIC cost function. These steps are given as follows: In the E-step, first we compute the expectation of the hidden data *Z*:(18)$$\begin{array}{cc} p(Z_{id}^{\left(j\right)}=1|w_{id};\Theta ,H) & =\frac{p\left(w_{id}|Z_{id}^{\left(j\right)};\Theta ,H\right)p\left(Z_{id}^{\left(j\right)}|\Theta ,H\right)}{\sum_{Z_{id}}p\left(w_{id}|Z_{id}^{\left(j\right)};\Theta ,H\right)p\left(Z_{id}^{\left(j\right)}|\Theta ,H\right)}\\ & =\frac{\alpha_{jd}v_{jd}\beta_{jn}^{u_{jw_{id}}}\beta_{0n}^{1-u_{jw_{id}}}}{\sum_{l=1}^{M}\alpha_{ld}v_{ld}\beta_{ln}^{u_{lw_{id}}}\beta_{0n}^{1-u_{lw_{id}}}}. \end{array}$$

With the expectation of the hidden data, we can compute the expected complete data log-likelihood using Equation (Equation 17). By replacing the log(p(D|H,Θ)) term in BIC with the expected complete data log-likelihood($E[L_{c}]$), we get the expected complete data BIC.

In the generalized M-step, based on the expectation of the complete data BIC we computed in the E-step, we update the model structure *H* and the model parameters Θ. First we optimize the model parameters given fixed model structure. Then we optimize the model structure given fixed model parameters, both steps taken to minimize the expected BIC. These steps are alternated until convergence.

When updating the model parameters, note that the only term in BIC that is related to the model parameters is the data likelihood term, so we can just maximize the expected complete data log-likelihood computed in the E-step to choose the model parameters. Taking those two constraints (Equations (Equation 2) and (Equation 3)) into consideration, we have our Lagrangian objective function.
(19)$$\begin{array}{cc} L & =\sum_{d=1}^{D}\sum_{i=1}^{L_{d}}\left(v_{jd}p\left(Z_{id}^{\left(j\right)}=1|w_{id};\Theta ,H\right)\left(log\left(\alpha_{jd}\right)+u_{jw_{id}}log\left(\beta_{jw_{id}}\right)+\left(1-u_{jw_{id}}\right)log\left(\beta_{0w_{id}}\right)\right)\right)\\ & -\sum_{d=1}^{D}\lambda_{d}\left(\sum_{j=1}^{M}\alpha_{jd}v_{jd}=1\right)-\sum_{j=1}^{M}\mu_{j}\left(\sum_{j=1}^{M}\left(u_{jn}\beta_{jn}+\left(1-u_{jn}\right)\beta_{0n}\right)-1\right), \end{array}$$
where $\mu_{j}$ and $\lambda_{d}$ are Lagrange multipliers. By computing the partial derivative of each parameter type and setting those derivatives to zero, we can get the optimized model parameters, satisfying necessary optimality conditions as:(20)$$\alpha_{jd}=\frac{\sum_{i=1}^{L_{d}}p\left(Z_{id}^{\left(j\right)}=1|w_{id};\Theta ,H\right)v_{jd}}{\sum_{l=1}^{M}\sum_{i=1}^{L_{d}}p\left(Z_{id}^{\left(l\right)}=1|w_{id};\Theta ,H\right)v_{ld}},j=1,\ldots ,M,d=1,\ldots ,D,$$
(21)$$\beta_{jn}=\frac{u_{jn}\sum_{d=1}^{D}\sum_{i=1:w_{id}=n}^{L_{d}}p\left(Z_{id}^{\left(j\right)}=1|w_{id};\Theta ,H\right)v_{jd}}{\mu_{j}},j=1,\ldots ,M,n=1,\ldots ,N.$$

We compute $\mu_{j}$ by multiplying both sides of Equation (Equation 21) by $u_{jn}$, summing over all *n*, and applying the distribution constrains on topic *j*. This gives:(22)$$\mu_{j}=\frac{\sum_{n=1}^{N}u_{jn}\sum_{d=1}^{D}\sum_{i=1:w_{id}=n}^{L_{d}}p\left(Z_{id}^{\left(j\right)}=1|w_{id};\Theta ,H\right)v_{jd}}{1-\sum_{j=1}^{M}\left(1-u_{jn}\right)\beta_{0n}},\forall j.$$

For the shared parameters, we only estimate them once via global frequency counts at initialization and hold them fixed during the GEM algorithm. That is, we set:(23)$$\beta_{0n}=\frac{\sum_{d=1}^{D}\sum_{i=1:w_{id}=n}1}{\sum_{d=1}^{D}L_{d}},\forall n.$$

When updating the model structure, we implement an iterative loop in which all the topic switches *u* are visited one by one. If the current change reduces BIC, we accept the change; otherwise we keep the switch unchanged. Note that in updating the word switches *v*, we update all the word switches for a single word jointly to see if it is optimal to choose all the switches related to a single word to be closed (i.e., to specify that this word is completely uninformative). This process is repeated over all the switches until there is no decrease in BIC or until we reach a pre-defined maximum number of iterations. We first update the word switches *u* until convergence; then we update the topic switches *v* until convergence. Then we go back to the E-step. Note that when updating the word switches, for each single search of all the switches related to one word, we have three configurations. All switches are closed, all switches are open, or some are closed and some are open. We compute the minimized BIC for each configuration, and then choose the configuration that has the lowest BIC value.

### 3.5. Selecting the Model Order

The optimization process discussed above is under the assumption that the model order is known. Model order selection is based on applying the optimization process for different model orders, in a top-down fashion. We initialize the model with a specific number of topics ($M_{max}$, chosen to upper bound the number of topics expected to be present in the corpus) and reduce the number by a predefined step Δ. For the model trained at each order, we remove the Δ topics with the smallest mass. This process is applied iteratively until the predefined minimum order is reached. We then retain the model (and model order) with the smallest BIC cost.

For our model, the only “hyper-parameters” are $M_{max}$ and Δ. We can expect the best performance by choosing Δ=1. For $M_{max}$, if this is set too small, it will underestimate the model order. If it is set very large, the learning and model order selection will require more computation. In principle, choosing any value of $M_{max}$ above the ground truth $M^{*}$ or the BIC minimizing $\hat{M}$ should be reasonable to find a good solution (the bottom line is that our method requires no true hyper-parameters—the only tradeoff is more computation by choosing Δ=1 and sufficiently large $M_{max}$).

## 4. Experiments and Results

In this section we compare the original PTM, LDA, and the new PTM method. All methods were used to solve the unsupervised density modeling problem, with no knowledge of class labels. Hence, the comparison of the methods is fair, and we evaluated multiple measures that assess the quality of the learned models as a function of the number of components in the model. None of the methods set hyper-parameters (except LDA, for which M was set to the the maximum)

Performance measurements: BIC was compared between the PTM model and our revised PTM model. For both models, the held-out log-likelihood and the class label purity were also compared. In [3] the performance of the PTM model and the LDA model were compared, and PTM was found to outperform LDA. Here we only include the label purity of the LDA model on different datasets.

When computing the held-out likelihood, we used the method described in [10,11] to compare model fitness on a held-out test set. We divided the documents in the test set into two parts: the observed part and the held-out part. First we computed the topic proportions based on the observed part, then we computed the held-out log-likelihood based on the held-out part:(24)$$\sum_{d=1}^{D_{test}}\sum_{i=1}^{L_{d}^{heldout}}log\left(\sum_{j=1}^{M}E_{q}\left[\alpha_{jd}\right]E_{q}\left[\beta_{jw_{id}}\right]\right).$$In our model $E_{q}\left[\alpha_{jd}\right]$ is directly the topic proportions $\alpha_{jd}$. $E_{q}\left[\beta_{jw_{id}}\right]$ is $u_{jn}\beta_{jn}+\left(1-u_{jn}\right)\beta_{0n}$.

We evaluated PTM, modified-PTM, and LDA on three datasets as next discussed.

### 4.1. Reuters-21578

The Reuters-21578 dataset is a collection of documents from Reuters news in 1987. There are in total 7674 documents from 35 categories. After stemming and stop word removal, there were 17,387 unique words. There were 5485 documents in the training set and 2189 documents in the test set.

### 4.2. 20-Newsgroups

The 20-Newsgroup dataset is a collection of 18,821 newsgroup documents from 20 classes. There were 53,976 words after stemming and stop word removal. It was split into a 11,293-document training set and a 7528-document test set.

### 4.3. Ohsumed

The Ohsumed dataset includes medical abstracts from the MeSH categories of the year 1991. It consists of 34,389 documents, each assigned to one or multiple labels of the 23 MeSH disease categories. Each document has on average 2.26 labels. The dataset was divided into 24,218 training and 10,171 test documents. There were 12,072 unique words in the corpus after applying standard stop word removal and stemming.

Note that for the Ohsumed dataset, each document may be associated with multiple labels. We computed the label purity for this dataset as follows: We first associated to each topic a multinomial distribution on the class labels. We learned these label distributions for each topic by frequency counting over the ground truth class labels of all documents, weighted by topic proportions:(25)$$p_{j}\left(c\right)=\frac{\sum_{d=1}^{D}\sum_{i=1:l_{id}=c}^{|C_{d}|}\alpha_{jd}v_{jd}}{\sum_{d=1}^{D}\sum_{i=1}^{|C_{d}|}\alpha_{jd}v_{jd}},\forall j,c,$$
where $p_{j}\left(c\right)$ is the class proportion for class *c*, for topic *j*. Here $l_{id}$ is the *i*-th class label in document *d* and $|C_{d}|$ is the number of class labels in document *d*. For labeling a text document, we then computed the probability of each class label based on the topic proportions in that document, that is, $\sum_{j=1}^{M}\alpha_{jd}p_{j}\left(c\right)$, and assigned the labels that had probability higher than a threshold value *T*:(26)$${\hat{C}}_{d}=\left\{c:\sum_{j=1}^{M}\alpha_{jd}p_{j}\left(c\right)>T\right\}.$$

We changed the threshold *T* and measured the area under the precision/recall curve (AUC). Note that here precision is the number of true discovered labels divided by the total number of ground-truth labels. Recall is the number of correctly classified labels divided by the total number of labels assigned to documents by our classifier.

### 4.4. Discussion

Our results on the three text corpora are shown in Figure 1, Figure 2 and Figure 3. Here we only include comparison with LDA on label purity. The comparison of LDA with the original PTM on held-out log-likelihood is in [3], and demonstrates that original PTM gives better results. Here, we see that the new PTM method convincingly outperformed original PTM (and hence LDA). Previous work showed that the original PTM method outperformed LDA with respect to two performance measurements: held-out log-likelihood and label purity [3]. This was attributed to the fact that PTM models are sparser than LDA models. However, the modified-PTM method dominated the original PTM (and LDA) with respect to all three of these measures, on all three datasets. Note in particular the large gain in held-out log-likelihood on Reuters-21578, in all measures on 20-Newsgroups, and with respect to BIC and held-out log-likelihood on Ohsumed.

The modified PTM minimizes BIC by selecting a much larger (richer) set of topics than the original PTM. This is achieved by the low description length penalty associated with making words completely uninformative (all closed), which leads to many more words being deemed completely uninformative compared to the original PTM. This is a key advantage of the new PTM method. The original PTM method offers no incentive within the BIC cost function to decide that a word is wholly uninformative about (irrelevant to) the topics that are present in the document. The new method provides substantial incentive for such determination with great reduction in model description complexity gained by such a choice. This allows the model to “afford” having many more topics than in the original PTM method. Especially, close to 80% of words were closed under modified PTM compared to approximately 40% for PTM for Reuters (at the selected model order), approximately 80% compared to approximately 50% are closed for 20-Newsgroups, and approximately 70% compared approximately to 30% were closed for Ohsumed. That is, modified PTM chose higher model orders (number of topics) with fewer topic-specific words and more wholly uninformative words than the original PTM. Choosing more topics was seen to yield better performance for several measures (label purity and held-out log-likelihood).

### 4.5. Computational Complexity

In both the PTM model and our modified model, we need to optimize over the parameters Θ and the model structure parameters {v,u}.

Consider learning the parameters Θ when M is fixed and with *D* documents. The computational complexity of the PTM model and our modified model are both $O(M_{D}DL_{D})$, where $M_{D}$ and $L_{D}$ are the number of topics present in a document and the length of a document, respectively. In LDA all topics are present in all documents, and the computational complexity is $O(MDL_{D})$—that is, the total computational complexity of the LDA model. However, in our model and the PTM model we also need to optimize over the model structure parameters {v,u}.

For the word switches $\left\{u_{jn}\right\}$, in the PTM model, updating them involves an iterative loop over all *N* words in M topics. Thus, the complexity is O(MN). In our modified PTM model for each of the *N* words, we need to check two more cases—thart is, case 1) and case 2) discussed before Equation (Equation 15). So, the computational complexity is O((M+2)N)=O(MN).

For the topic switches $\left\{v_{jd}\right\}$, both our model and the PTM model need computations of order $O(M_{D}L_{D})$ to update each switch $v_{jd}$. We have in total MD switches to be updated. So the computational complexity of updating the topic switches is $O(MM_{D}DL_{D})$ for both PTM and our modified model. The computational complexity comparison is shown in Table 1.

We recorded the execution time for the PTM model and the revised PTM model on each dataset (in Table 1). We ran the experiment on a machine with an Intel Core i5, 2.3 GHz processor. The execution time of our method was slightly less than that of the PTM model, which may be because in our modified model, for the word switches related to a single word, when it comes to the point that all the switches are closed, it is very likely that no update will be done in the future iterations. We did an experiment with the LDA model using the Python package, but our modified PTM model is implemented in C. So, the comparison between our model and LDA may be unfair. A comparison of the execution time between LDA and PTM models is reported in [3].

### 4.6. Limitations of Our Work

One limitation of our model is that the computational complexity is higher than for the LDA model. Another limitation is the highly non-convex optimization, with discrete and continuous parameters and no guarantee of finding the global minimum.

## 5. Conclusions

In this paper, we proposed two improvements on the PTM method. One is improving the modeling by giving a cheaper expression to describe the wholly uninformative words. The other is improving the learning—we optimized all the parameters related to a single word under each topic, which encourages a sparser model. Our improvements led to consistent and substantial gains in modeling accuracy with respect to multiple performance measures, compared with both the original PTM method and the well-known LDA model. While we demonstrated significant gains on three document corpora, there is increased computational complexity for our method compared to LDA. Future work could aim to develop a fully Bayesian version of our approach, based on posteriors on topic and word switches, rather than on deterministic (binary) switch parameters. Future work could also investigate extensions that model word order dependency, but somehow in a low-complexity fashion.

## Figures and Tables

**Figure 1 entropy-22-00326-f001:**
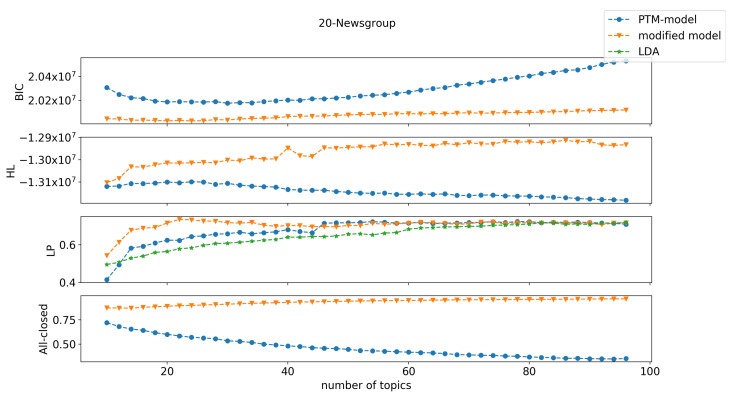
Performance comparison between the original parsimonious topic model (PTM), our modified PTM, and latent Dirichlet allocation (LDA) on the Reuters-21578 dataset. The measurements were the Bayesian information criterion (BIC), held-out log-likelihood (HL), label purity (LP), and the proportion of words with all-closed word switches (All-closed). LDA is only shown for label purity.

**Figure 2 entropy-22-00326-f002:**
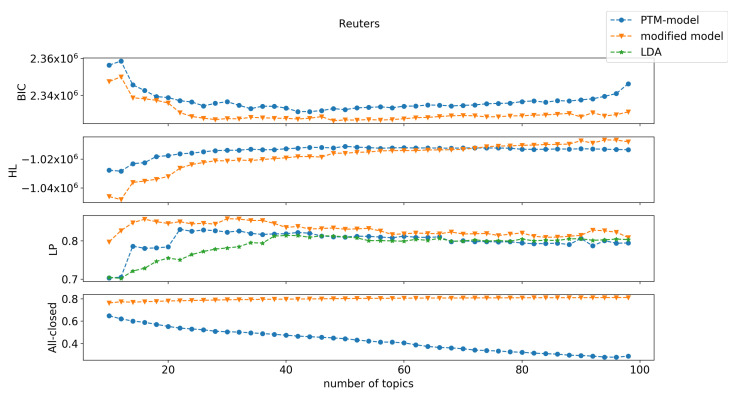
Performance comparison between the original PTM, our modified PTM, and LDA on the 20-Newsgroup dataset. The measurements were BIC, held-out log-likelihood (HL), label purity (LP), and the proportion of the words with all-closed word switches (All-closed). LDA is only shown for label purity.

**Figure 3 entropy-22-00326-f003:**
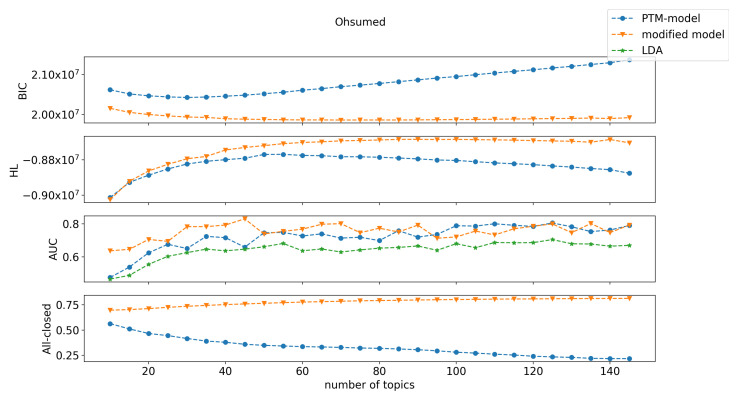
Performance comparison between the original PTM, our modified PTM, and LDA on the Ohsumed dataset. The measurements were BIC, held-out log-likelihood (HL), AUC, and the proportion of the words with all-closed word switches (All-closed). LDA is only shown for label purity.

**Table 1 entropy-22-00326-t001:** Comparison of thecomputational complexity and execution time of different models.

	Modified PTM	PTM	LDA
Complexity	$O(MM_{D}DL_{D}+MN)$	$O(MM_{D}DL_{D}+MN)$	$O(MDL_{D})$
**Execution Time (min)**			
20-Newsgroup	387	493	
Reuters-21578	39	44	
Ohsumed	424	659

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
