# Peer review of "Improved Parsimonious Topic Modeling Based on the Bayesian Information Criterion"

_entropy, 2020, doi:10.3390/e22030326_

Round 1

Reviewer 1 Report

1. The sentence in line 88

"Here vjd is the topic switch that indicates whether topic j is present in document d; only if vjd = 1  is ajd a model parameter. ujn is the word switch that indicates if word n is topic-specific under topic  j."

is awkward because of some typo. Please correct it.

2. In line 83 to 85, for describing pmfs, define \alpha_{jd}, v_{jd},

   \beta_{jn}, \beta_{0n} before the line 83 instead of doing this in line 93.

The paper should be carefully revised from the beginning to the end..

Author Response

Dear reviewer

We are very grateful to your comments for the manuscript. We modified the paper accordingly, the details are as follows:

1. The sentence in line 88 of the original paper has been modified and improved.

2. We moved the definitions of  \alpha_{jd}, v_{jd}, \beta_{jn} and \beta_{0n} to the line following the pmf (lines 83 to 85)

We also revised the entire paper to improve language, readability, and to include more results discussion, as well as discussion of limitations of our method. We also improved the figures and now include computational complexity analysis.

Thanks again for you suggestions!

Reviewer 2 Report

This is a carefully written paper, however I was not able to determine the total correctness of all of the equations and hence I assume that these are correct. The application to three different types of data sets makes the modeling more convincing to the reader. However Figures 1, 2 and 3 on pages 8 and 9 are too small and difficult to discern the differences both within and between each of the 4 sections with the vertical axes names difficult to read. More verbal discussion should be provided in section 4.4 to distinguish the difference of the results of Figure 1 versus Figure 2, Figure 1 versus Figure 3, and Figure 2 versus Figure 3. The first sentence of Section 4.4 is missing period at end of sentence ending with "Figs. 1-3." with removal of extra space before "1-3".

The conclusion in section 5 consists of only one sentence and should be more.

Author Response

Dear reviewer:

Thank you for your kind comments. We revised the manuscript accordingly. Here below is our description on revisions according to the comments.

We modified the figures by making them larger, using larger font size, and using symbols to distinguish different methods. We also added more description in the caption of the figures to make the names of four measurements in the vertical axes clear. Finally we explain in the text and figure caption that LDA is only evaluated in this paper on the label purity measure.

We added more discussion in section 4.4 about the comparison results in the three figures.

The conclusion section has now been expanded. Moreover, we also included a section in limitations of our method.

Reviewer 3 Report

  My comments are as follows:

  • The computational complexity of the proposed solver should be discussed. Also, the experiment time for those who are interested in execution time. I suggest using the big oh notation.

  • To have an unbiased view in the paper, there should be some discussions on the limitations the solver

  • Analysis of the results is missing in the paper. There is a big gap between the results and conclusion. There should be the result analysis between these two sections. After comparing the methods, you have to be able to analyse the results and relate them to the structure of solver. It would be interesting to have your thoughts on why the method works that way? What is the role of the operators proposed in this work? Such analyses would be the core of your work where you prove your understanding of the reason behind the results. You can also link the findings to the hypotheses of the paper. Long story short, this paper requires a very deep analysis from different perspectives

  • How do you ensure that the comparison between the methods is fair?

  • The solver might be sensitive to the values of its main controlling parameter. How did you tune the parameters?

Author Response

Dear reviewer

We are very grateful to your comments for the manuscript. We modified the paper accordingly. Here below is our description on revisions according to the comments.

1. We created a new section (4.5 Computational Complexity). We included the discussion of the Computational Complexity for our model, PTM model, and the LDA model and the execution time of our model and PTM model on different datasets.

2. A new section is added (4.6 Limitations of our work). Two limitations of our work are as follows:

    No guarantee of global optimum, for a complicated, non convex optimization problem.

    The time complexity is higher than the LDA model

3. We discussed the original PTM and LDA and why PTM achieved gains over LDA; then explained why the new method achieves even more gains (in section 4.4).

4. We emphasized at the beginning of section 4 that all methods  are completely unsupervised; no method exploits any more information than any other. So the comparisons are fair.

5. We analyzed the hyper-parameters starting from line 207. There are two “hyper-parameters”: M_{max} and \Delta. We can expect best performance by choosing \Delta = 1. And for M_max, if this is set too small, it will underestimate the model order. If this is set very large, the learning and model order selection will require more computation. In principle, choosing any value of M_{max} above the ground truth M^* or the BIC minimizing \hat{M} should be reasonable. So M_{max} is also not a hyper-parameter that needs to be very carefully chosen to achieve strong results—it should just not be chosen too small.

Thanks again for your kind comments.